# Transgenic expression of *cif* genes from *Wolbachia* strain *w*AlbB recapitulates cytoplasmic incompatibility in *Aedes aegypti*

Cameron J. McNamara[1], Thomas H. Ant[1], Tim Harvey-Samuel [2], Helen White-Cooper[3], Julien Martinez [1], Luke Alphey [2,4] & Steven P. Sinkins [1] ✉

The endosymbiotic bacteria *Wolbachia* can invade insect populations by modifying host reproduction through cytoplasmic incompatibility (CI), an effect that results in embryonic lethality when *Wolbachia*-carrying males mate with *Wolbachia*-free females. Here we describe a transgenic system for recreating CI in the major arbovirus vector *Aedes aegypti* using CI factor (*cif*) genes from *w*AlbB, a *Wolbachia* strain currently being deployed to reduce dengue transmission. CI-like sterility is induced when *cifA* and *cifB* are co-expressed in testes; this sterility is rescued by maternal *cifA* expression, thereby reproducing the pattern of *Wolbachia*-induced CI. Expression of *cifB* alone is associated with extensive DNA damage and disrupted spermatogenesis. The strength of rescue by maternal *cifA* expression is dependent on the comparative levels of *cifA*/*cifB* expression in males. These findings are consistent with CifB acting as a toxin and CifA as an antitoxin, with CifA attenuating CifB toxicity in both the male germline and in developing embryos. These findings provide important insights into the interactions between *cif* genes and their mechanism of activity and provide a foundation for the building of a *cif* gene-based drive system in *Ae. aegypti*.

Many strains of the maternally transmitted intracellular bacteria *Wolbachia* can invade arthropod populations and remain at high frequency through modifications of host reproduction. Most commonly this occurs via a mechanism known as cytoplasmic incompatibility (CI), whereby modifications of the paternal chromatin result in embryonic lethality when *Wolbachia*-carrying males mate with *Wolbachia*-free females. If the females carry a compatible *Wolbachia* strain, CI-induced lethality is rescued, which provides *Wolbachia*-carrying females with a relative fitness advantage. The genetic basis for incompatibility stems from two syntenic and co-diverging genes which are broadly referred to as the CI factors, *cifA* and *cifB*[1,2]. These *cif* gene sequences can be highly divergent and homologues are sorted into five phylogenetic groups (Types I-V)[3]. *Wolbachia* strains can possess multiple pairs of similar or divergent *cif* gene types, which can result in complex patterns of mating incompatibility between strains[4,5].

CI has been recapitulated using transgenic systems expressing the *cif* genes and CI induction and rescue have been attributed to CifB and CifA respectively[2,6–14]. However, for some host species and/or *cif* homologue combinations, the simultaneous co-expression of both *cifA* and *cifB* was found to be necessary for CI induction[1,6,9]. This finding led to a proposed "two-by-one" genetic model of CI, as both CifA and CifB were needed for CI induction while only CifA was required for rescue[9] – although there are a minority of cases where *cifB* expression alone has been able to induce rescuable CI[10,11]. Despite the frequent requirement for *cifA* co-expression in achieving CI induction, its function in generating rescuable sterility is unknown.

[1]MRC—University of Glasgow Centre for Virus Research, 464 Bearsden Road, Glasgow G61 1QH, UK. [2]Arthropod Genetics, The Pirbright Institute, Ash Road, Pirbright, Surrey GU24 ONF, UK. [3]Molecular Biosciences Division, Cardiff University, Cardiff CF10 3AX, UK. [4]The Department of Biology, University of York, Wentworth Way, York YO10 5DD, UK. ✉e-mail: steven.sinkins@glasgow.ac.uk

There are currently two competing theories for how CI functions: the toxin-antidote (TA)[15] and host-modification (HM)[16] models. A key difference between these models concerns at which stage the modification(s) associated with CI induction occur. The TA model predicts that sperm-deposited CifB modifies the paternal chromatin in fertilised embryos unless inhibited by maternally-inherited CifA present in the zygote[8,12,15,17]. The CifA "antidote" is expected to neutralise CifB toxicity through the specific binding of cognate Cif pairs, which explains why incompatibility arises between *Wolbachia* strains encoding different Cif homologues. As the TA model predicts that for rescuable CI induction the CifB modification(s) should occur in the embryo, a potential role of CifA in CI induction is the prevention of premature modifications during spermatogenesis. Alternatively, the HM model predicts that CifB-mediated modification(s) occur during spermatogenesis[16,18], with CifA acting as a co-factor. This model does not necessitate the packaging of either Cif peptide into sperm, and posits that rescue is attributed to either CifA-mediated reversal of the paternal chromatin modifications or compensatory modifications of the maternal chromatin. However, the HM model provides a less parsimonious explanation for the complex and diverse patterns of compatibility between *Wolbachia* strains observed in nature, as this would require a large number of host targets[19].

Despite the high prevalence of *Wolbachia* infections in insect host species, a native strain of the bacterium has not been found in the major arbovirus mosquito vector *Aedes aegypti*[20]. *Wolbachia* strains introduced into *Ae. aegypti* from other insect host species have resulted in the generation of *Ae. aegypti* lines that exhibit strong CI[21–24] and block the transmission of some positive-strand RNA viruses of significant public health concern[25], such as dengue virus. Release of *Wolbachia*-carrying mosquitoes of both sexes can result in the spread and stable fixation of the bacterium in a population. The "replacement" strategy aims to spread *Wolbachia* strains that block the transmission of arboviruses and has so far focused on two strains *w*Mel (native to *Drosophila melanogaster*) and *w*AlbB (native to *Ae. albopictus*). Releases of *w*Mel-carrying *Ae. aegypti* in several countries by the World Mosquito Program (WMP)[26–28] and releases of *w*AlbB-carrying *Ae. aegypti* in areas of Greater Kuala Lumpur, Malaysia[29], have both led to significant decreases in dengue transmission. Alternatively, inundative male-only releases of *w*AlbB-carriers has successfully reduced vector population sizes by reducing reproductive capacity through incompatible mating[30–33]. *Wolbachia*-induced CI is fundamental to each of these approaches, but is as yet incompletely understood.

The *w*AlbB genome encodes two sets of cognate *cif* genes of different types (Type III and IV)[34]. Despite significant sequence divergence, both sets possess the same putative protein domains and are assumed to be active since they do not contain predicted loss-of-function mutations; however, to date there have been no functional studies on them. CI has been successfully recreated using another Type IV *cif* gene set (from the *w*Pip strain, derived from the mosquito *Culex pipiens*) in transgenic *D. melanogaster*[7,10]. To build a transgenic system to synthetically recapitulate CI induction and rescue in a major arbovirus vector, we generated an *Ae. aegypti* germline expression system utilising the Type IV *cif* genes from *w*AlbB (*cifA*$_{w\text{AlbB(TIV)}}$ and *cifB*$_{w\text{AlbB(TIV)}}$). The findings of this study provide insights into the role of CifA in the male germline and lays the groundwork for the construction of a transgenic system capable of inheritance at greater than Mendelian ratios; such a synthetic *cif* gene drive system could be used for the delivery of antiviral effectors in *Ae. aegypti*.

## Results

### *cifA*$_{w\text{AlbB(TIV)}}$/*cifB*$_{w\text{AlbB(TIV)}}$ co-expression in testes is required for CI induction

A *piggyBac*-based transgene construct comprising *cifB*$_{w\text{AlbB(TIV)}}$ transcriptionally regulated by the testis-specific *Beta-2-tubulin* (*β2t*) promoter[35] (Fig. 1a) was generated and microinjected into pre-

blastoderm *Ae. aegypti* embryos, resulting in two independent genomic insertion lines (B1 and B2). Crosses between *β2t-cifB*$_{w\text{AlbB(TIV)}}$ males and wild-type females resulted in complete sterility, and this was not rescuable by *w*AlbB *Wolbachia*-carrying females (Fig. 1b). Although mating was visually confirmed, dissection and microscopic analysis of the female spermathecae (sperm storage organs) revealed no sperm transfer (Fig. 1b). Dissection of the testes and seminal vesicles from *β2t-cifB*$_{w\text{AlbB(TIV)}}$ males showed a disruption in the production of mature spermatozoa (Supplementary Movie 1).

We hypothesised that testis-specific expression of *cifA*$_{w\text{AlbB(TIV)}}$ might restrict the toxicity of CifB$_{w\text{AlbB(TIV)}}$, resulting in restored spermatogenesis. A *piggyBac*-based transgene construct comprising the *β2t* promoter driving *cifA*$_{w\text{AlbB(TIV)}}$ expression (Fig. 1a) was generated and microinjected into *Ae. aegypti* embryos, resulting in two independent genomic *β2t-cifA*$_{w\text{AlbB(TIV)}}$ insertion lines (A1 and A2). Heterozygous males from each line were crossed with either wild-type or *w*AlbB *Wolbachia*-carrying females, and effects on embryo viability were assessed. No reduction in median viability was observed, indicating no effect of *β2t*-expressed *cifA*$_{w\text{AlbB(TIV)}}$ on male fertility (Fig. 1b).

To test whether *cifA*$_{w\text{AlbB(TIV)}}$ attenuates *cifB*$_{w\text{AlbB(TIV)}}$ toxicity in testes, sub-lines heterozygous for both *β2t-cifA*$_{w\text{AlbB(TIV)}}$ and *β2t-cifB*$_{w\text{AlbB(TIV)}}$ were generated and males from each sub-line were crossed with either wild-type or *w*AlbB *Wolbachia*-carrying females. These females were bloodfed and allowed to oviposit, after which spermathecae were dissected and the frequency of sperm transfer was assessed (Fig. 1b). Co-expression of *β2t-cifA*$_{w\text{AlbB(TIV)}}$ was found to rescue the *β2t-cifB*$_{w\text{AlbB(TIV)}}$ mediated disruption of spermatogenesis, although this rescue was dependent on the comparative expression levels of *cifA*$_{w\text{AlbB(TIV)}}$ and *cifB*$_{w\text{AlbB(TIV)}}$. RT-qPCR targeting *cifB*$_{w\text{AlbB(TIV)}}$ in testes from the *β2t-cifB*$_{w\text{AlbB(TIV)}}$ insertion lines showed significant between-line variation in expression – likely due to positional effects of the transgene in the different genomic loci (Fig. 1c). When the mean relative expression of *cifA*$_{w\text{AlbB(TIV)}}$ was comparable to that of *cifB*$_{w\text{AlbB(TIV)}}$ in the testes (A1/A2;B1), all mated female spermathecae contained sperm (Fig. 1b). However, when the mean relative expression of *cifA*$_{w\text{AlbB(TIV)}}$ was lower than that of *cifB*$_{w\text{AlbB(TIV)}}$ in the testes (A1/A2;B2), the majority of mated female spermathecae contained no sperm (Fig. 1b). Despite the successful production and transfer of sperm to wild-type females by males combining *cifB*$_{w\text{AlbB(TIV)}}$ B1 with either the *cifA*$_{w\text{AlbB(TIV)}}$ A1 or A2 insertions, all the crosses resulted in inviable embryos (Fig. 1b). However, if the females carried *w*AlbB *Wolbachia* there was a significant increase in the viability rate of the offspring (median of 19.5% and 44.5% when crossed with A1;B1 and A2;B1 males respectively), indicating that at least part of the sterility observed for *β2t-cifA*$_{w\text{AlbB(TIV)}}$;*cifB*$_{w\text{AlbB(TIV)}}$ males was a result of the same *cif* gene mechanism that occurs in canonical CI induction (Fig. 1b).

As *cifB*$_{w\text{AlbB(TIV)}}$ encodes two PD-(D/E)XK nuclease domains that are predicted to be functional due to the presence of known catalytic residues[3], it was hypothesised that the spermless phenotype might be the result of arrested sperm development due to DNA damage arising from CifB nuclease activity. TUNEL (terminal deoxynucleotidyl transferase dUTP nick-end labelling) assays of testes squashes from *β2t-cifB*$_{w\text{AlbB(TIV)}}$ males revealed considerable DNA damage in comparison to wild-type or *β2t-cifA*$_{w\text{AlbB(TIV)}}$ testes (Fig. 1d). As DNA breaks also occur during apoptosis, this assay does not discriminate whether the DNA damage observed is the direct result of CifB$_{w\text{AlbB(TIV)}}$ nuclease activity or whether CifB$_{w\text{AlbB(TIV)}}$ induces apoptosis through another mechanism.

### Paternal Cif dosage affects maternal rescue

Placement of a T2A self-cleaving peptide sequence between two adjacent protein coding sequences allows their transcription as a single mRNA but their translation as two independent peptides at near equimolar quantities through ribosomal skipping[36]. As comparable

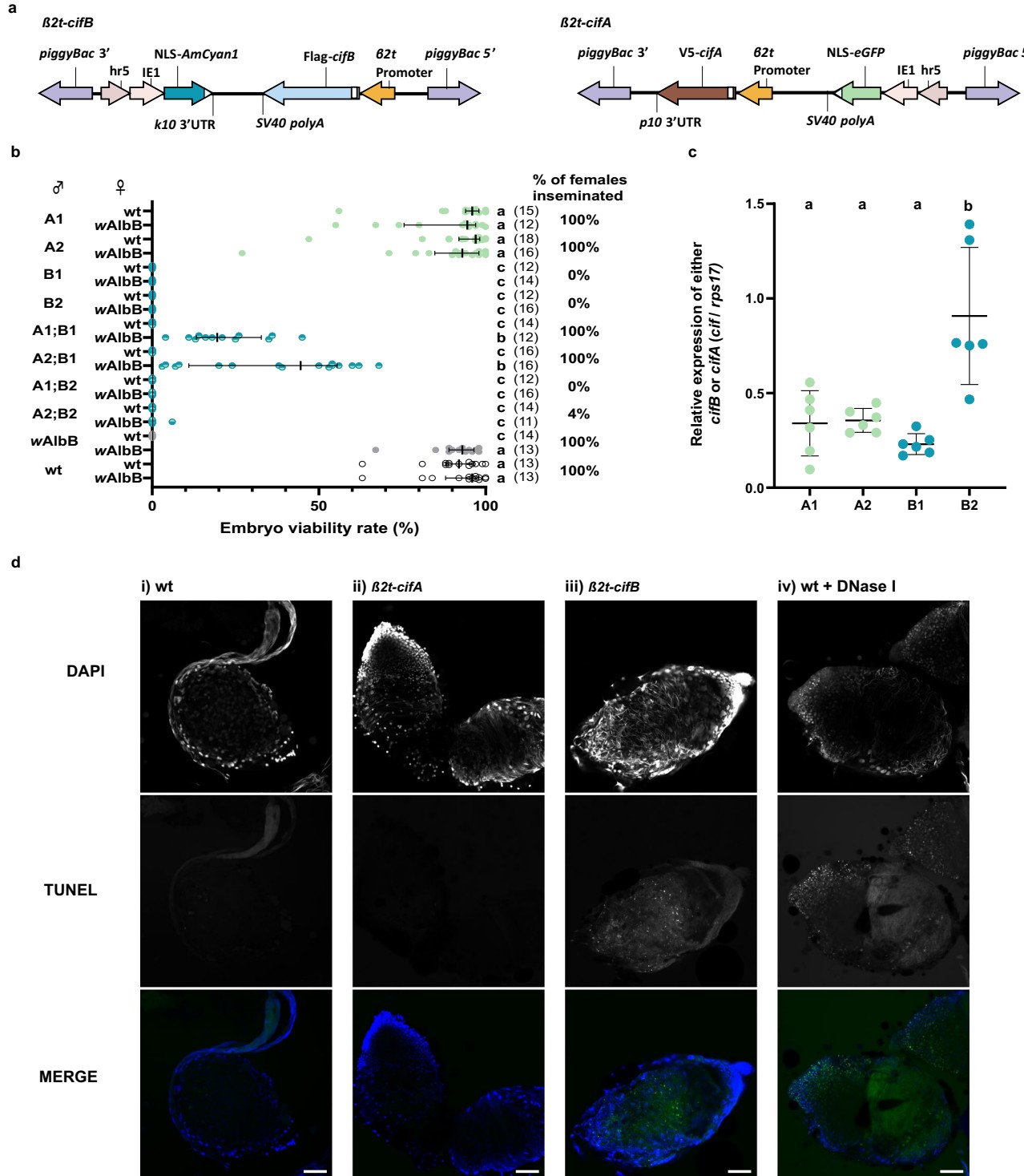

**Fig. 1 | CifA attenuates the toxicity of CifB during spermatogenesis resulting in rescuable CI. a** Construct design of *ß2t-cifB* and *ß2t-cifA*. **b** Expression of *cifB* alone from either genomic insertion site (B1 or B2) impaired sperm production (indicated by the percentage of females inseminated in each cross) rendering males infertile, while expression of *cifA* from either genomic insertion site (A1 or A2) did not affect male fertility. Depending on the combination of genomic insertion sites, co-expression of *cifA;cifB* could rescue sperm production and induce sterility when crossed with wild-type (wt) females, which was rescued in crosses with *w*AlbB-carrying females. Lines denote median, error bars interquartile ranges, and numbers in parentheses the *n*. Letters indicate significant differences with an α = 0.05 calculated by a Kruskal–Wallis test: H = 261.7, *P* < 0.0001, d.f. = 19 followed by a two-stage linear step-up procedure of Benjamini, Krieger, and Yekutieli to correct for multiple comparisons, individual *P*-values are listed in the Source Data file. **c** The relative expression of either *cifA* or *cifB* relative to housekeeping gene *rps17* in pooled adult testes (mean and s.d. are shown, *n* = 5 for each group). Letters indicate significant differences with an α = 0.05 calculated by a one-way ANOVA and Tukey's post-hoc multiple pairwise comparisons test. **d** TUNEL staining on testes squashes from >5 day old males. DNA breaks labelled by TUNEL staining were not observed in i) wt or ii) *ß2t-cifA* testes, whereas an abundance of DNA breaks was observed in iii) *ß2t-cifB* and iv) wt testes treated with DNase I. DNA was labelled with DAPI stain. Scale bar, 50 μm. Source data are provided as a Source Data file.

$cifA_{w\text{AlbB(TIV)}}/cifB_{w\text{AlbB(TIV)}}$ expression levels resulted in rescuable CI induction when the genes were transcribed from different genomic loci, we generated $\beta2t$-$cifA_{w\text{AlbB(TIV)}}$-T2A-$cifB_{w\text{AlbB(TIV)}}$ (hereafter $\beta2t$-$cifA_{w\text{AlbB(TIV)}}$-$cifB_{w\text{AlbB(TIV)}}$) lines to test the effect of expressing the $cif$ genes from the same locus (Fig. 2a). Males carrying a single $\beta2t$-$cifA_{w\text{AlbB(TIV)}}$-$cifB_{w\text{AlbB(TIV)}}$ insertion were found to produce mature sperm and induce complete sterility when crossed with wild-type females (median of 0% viable embryos) (Fig. 2b). However, the level of rescue was low when these males were crossed with $w$AlbB *Wolbachia*-carrying females (median of 6% viable embryos) (Fig. 2b). This result was similar for all four independent $\beta2t$-$cifA_{w\text{AlbB(TIV)}}$-$cifB_{w\text{AlbB(TIV)}}$ genomic insertion lines generated (Supplementary Fig. 1).

As the comparative expression levels of the $cif$ genes was seen to affect the viability of sperm (Fig. 1b), it was hypothesised that increasing the expression of $cifA_{w\text{AlbB(TIV)}}$ relative to that of $cifB_{w\text{AlbB(TIV)}}$ would further inhibit sperm damage and improve the rescue capacity of $w$AlbB-carrying females. To increase the paternal dosage of CifA the previously described $\beta2t$-$cifA_{w\text{AlbB(TIV)}}$ line (A2) was crossed into the $\beta2t$-$cifA_{w\text{AlbB(TIV)}}$-$cifB_{w\text{AlbB(TIV)}}$ line. The penetrance of sterility induced by heterozygous $\beta2t$-$cifA_{w\text{AlbB(TIV)}}$;$\beta2t$-$cifA_{w\text{AlbB(TIV)}}$-$cifB_{w\text{AlbB(TIV)}}$ males was not attenuated when crossed with wild-type females (median of 0% viable embryos) (Fig. 2b). However, the higher dosage of $CifA_{w\text{AlbB(TIV)}}$ was found to significantly increase the rescue capability of $w$AlbB *Wolbachia*-carrying females to a level comparable to that of $w$AlbB *Wolbachia*-carrying male and female control crosses

(median of 75% and 86% viable embryos respectively) (Fig. 2b). To investigate the effect that lowering the overall levels of the Cif peptides in the testes had on CI penetrance and rescue capacity, a transgene construct was generated that swapped the $\beta2t$ promoter with the testes-specific *matotopetli* (*topi*)(AAEL023352) promoter (Fig. 2a), which is a weaker promoter than $\beta2t$ (Fig. 2c). In contrast to $\beta2t$-mediated $cifA_{w\text{AlbB(TIV)}}$-$cifB_{w\text{AlbB(TIV)}}$ expression, *topi*-driven $cifA_{w\text{AlbB(TIV)}}$-$cifB_{w\text{AlbB(TIV)}}$ expression resulted in full rescue by $w$AlbB *Wolbachia*-carrying females (Fig. 2b, c and Supplementary Fig. 1).

The toxicity of CifB peptides encoding functional nuclease domains (Types II-V) is likely to be dependent on nuclease catalytic activity; mutation of key catalytic residues in these domains ablates toxicity in transgenic yeast[1,7,10] and reduces CI penetrance in *Drosophila*[7]. However, it is not known if this activity occurs in the male reproductive tissues. To determine whether $cifA/B_{w\text{AlbB(TIV)}}$ co-expression resulted in DNA breaks in cells undergoing spermatogenesis, TUNEL assays were performed on testes dissected from $\beta2t$-$cifA_{w\text{AlbB(TIV)}}$-$cifB_{w\text{AlbB(TIV)}}$ males. Despite the production of mature spermatozoa, DNA breaks were still observed in $\beta2t$-$cifA_{w\text{AlbB(TIV)}}$-$cifB_{w\text{AlbB(TIV)}}$ testes (Fig. 3). However, no DNA breaks were observed in testes dissected from either the $\beta2t$-$cifA_{w\text{AlbB(TIV)}}$;$\beta2t$-$cifA_{w\text{AlbB(TIV)}}$-$cifB_{w\text{AlbB(TIV)}}$ or *topi*-$cifA_{w\text{AlbB(TIV)}}$-$cifB_{w\text{AlbB(TIV)}}$ lines (Fig. 3). As males from both these lines can induce complete CI (Fig. 2b), it suggests that inducing DNA breaks during spermatogenesis is not required for the induction of CI. Furthermore, no DNA breaks were observed in $w$AlbB

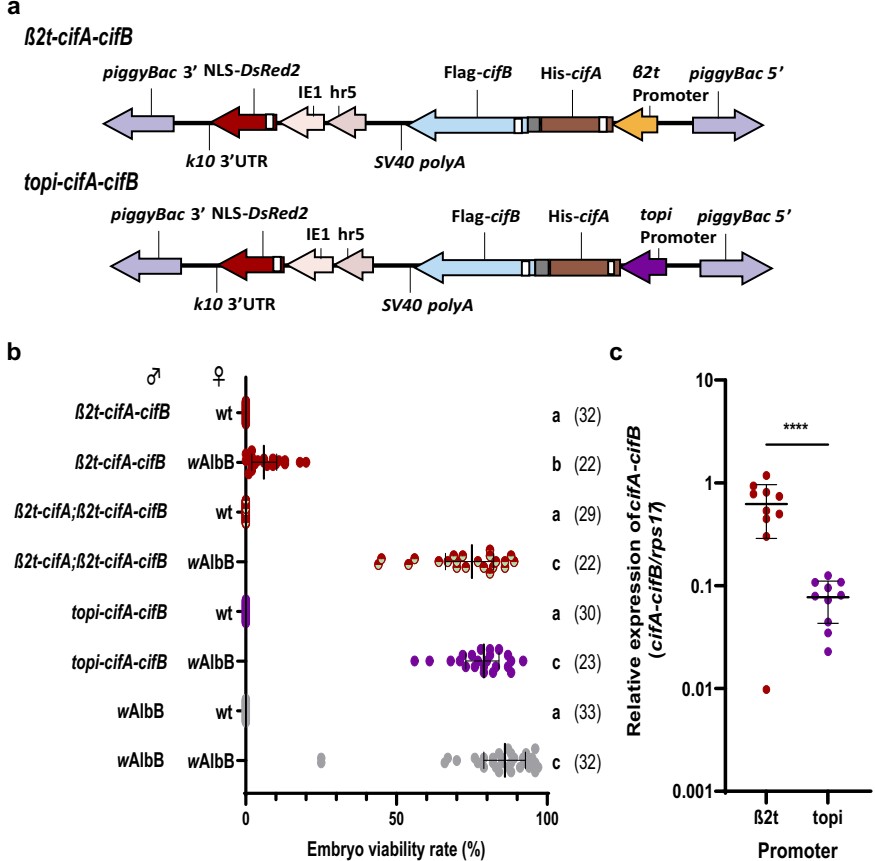

**Fig. 2 | The paternal dosage of Cifs affects the capacity of maternal CifA to rescue incompatibility. a** Construct design of *β2t-cifA-cifB* and *topi-cifA-cifB*. **b** Increasing the relative expression of *cifA* in comparison to *cifB* (*β2t-cifA;β2t-cifA-cifB*) or lowering the expression of both *cif* genes (*topi-cifA-cifB*) did not affect CI penetrance when males were crossed with wild-type (wt) females. However, it did improve the capacity of $w$AlbB-carrying females to rescue CI. Lines denote median and error bars interquartile ranges, numbers in parentheses denote the *n*. Letters indicate significant differences with an α = 0.05 calculated by a Kruskal–Wallis test: H = 209.7, $P < 0.0001$, d.f. = 7, followed by a two-stage linear step-up procedure of Benjamini, Krieger, and Yekutieli to correct for multiple comparisons, individual *P*-values are listed in the Source Data file. **c** The relative expression of the *cifA-cifB* transcript was significantly lower when the expression was regulated by the *topi* promoter (two-tailed unpaired *t*-test $p < 0.0001$, mean and s.d. are shown, $n = 10$ for both groups). Source data are provided as a Source Data file.

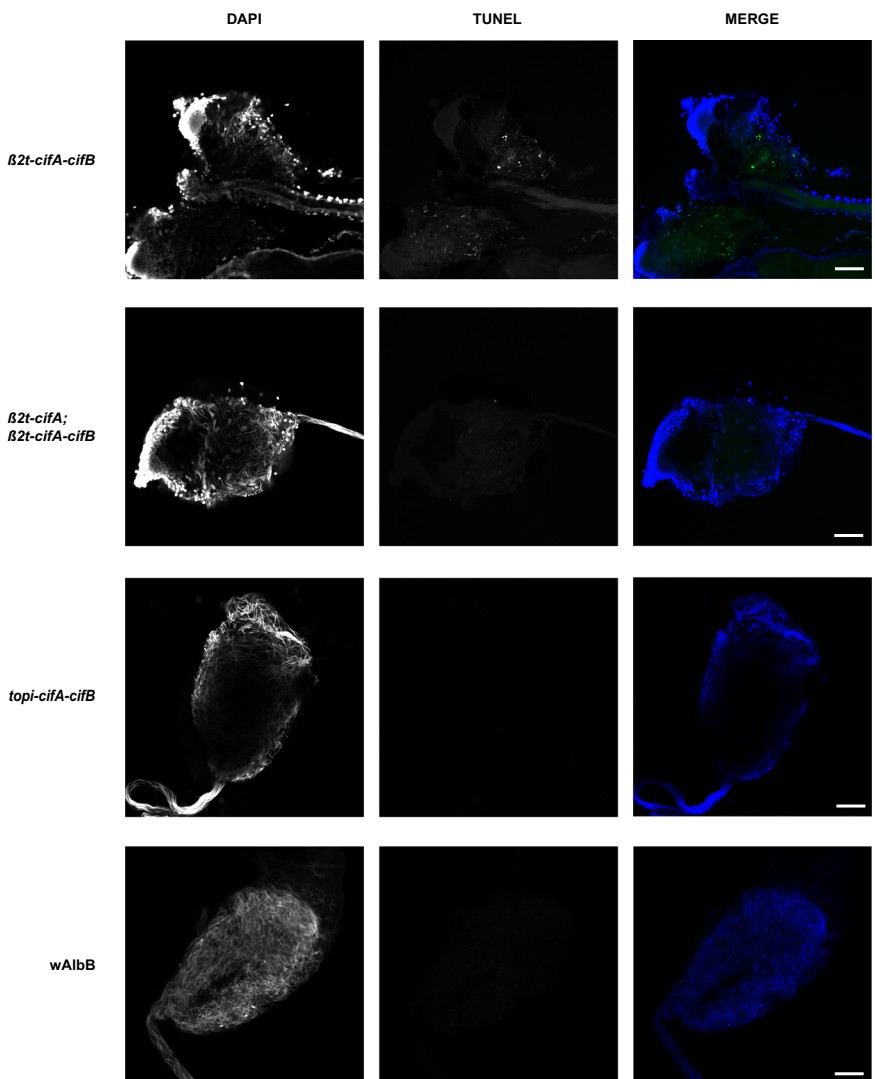

|  | DAPI | TUNEL | MERGE |
| --- | --- | --- | --- |
| *ß2t-cifA-cifB* | | | |
| *ß2t-cifA;*<br>*ß2t-cifA-cifB* | | | |
| *topi-cifA-cifB* | | | |
| *wAlbB* | | | |

**Fig. 3 | CifB-induced DNA damage is inhibited during spermatogenesis.** TUNEL assays on 5-day old *Ae. aegypti* dissected reproductive tissues revealed DNA breaks occurring in *β2t-cifA-cifB* testes which was inhibited by additional expression of *cifA* (*β2t-cifA;β2t-cifA-cifB*) whilst lowering the overall dosage of the Cifs (*topi-cifA-cifB*) also resulted in a lack of DNA breaks. Although *w*AlbB is predicted to produce two sets of nuclease CifB peptides no DNA damage was not observed in *w*AlbB-carrying testes. Scale bar, 50 μm.

*Wolbachia*-carrying testes despite *w*AlbB encoding two Types of CifB peptide with expected nuclease activity (Fig. 3).

### Maternal transgenic expression of $cifA_{w\mathrm{AlbB(TIV)}}$ rescues transgenic CifB$_{w\mathrm{AlbB(TIV)}}$-induced sterility

Now that CI induction through transgenic expression of *cifA/B*$_{w\mathrm{AlbB(TIV)}}$ had been demonstrated, a line that expressed *cifA*$_{w\mathrm{AlbB(TIV)}}$ in the female germline was generated to determine whether both CI induction and rescue could be achieved in a transgenic *Ae. aegypti* system. The *Ae. aegypti exuperantia* (*exu*) promoter was selected to drive *cifA*$_{w\mathrm{AlbB(TIV)}}$ (Fig. 4a) as it promotes high expression in the ovaries following a bloodmeal and deposition of the gene product in oocytes[37]. Because *w*AlbB encodes two-sets of incompatible *cif* gene pairs it was not expected that maternal transgenic expression of *cifA*$_{w\mathrm{AlbB(TIV)}}$ would rescue CI induced by *Wolbachia* carrying males. As expected, when heterozygous females from the resulting *exu-cifA*$_{w\mathrm{AlbB(TIV)}}$ line were crossed with *w*AlbB *Wolbachia*-carrying males the median embryo viability rate was not increased from 0% (Fig. 4b). However, when crossed with *β2t-cifA*$_{w\mathrm{AlbB(TIV)}}$-*cifB*$_{w\mathrm{AlbB(TIV)}}$ males, the embryo viability rate was significantly increased (median of 1.5% viable embryos) compared with wild-type females, indicating partial rescue

(Fig. 4b). Similar to *w*AlbB *Wolbachia*-carrying females the rescue capacity of *exu-cifA*$_{w\mathrm{AlbB(TIV)}}$ females was increased when crossed with either *β2t-cifA*$_{w\mathrm{AlbB(TIV)}}$;*β2t-cifA*$_{w\mathrm{AlbB(TIV)}}$-*cifB*$_{w\mathrm{AlbB(TIV)}}$ or *β2t-cifA*$_{w\mathrm{AlbB(TIV)}}$;*topi-cifA*$_{w\mathrm{AlbB(TIV)}}$-*cifB*$_{w\mathrm{AlbB(TIV)}}$ males. However, the rescue provided by maternal *cifA*$_{w\mathrm{AlbB(TIV)}}$ expression was lower (median of 28% and 24% viable embryos respectively) (Fig. 4b) than provided by the presence of *w*AlbB *Wolbachia* (median of 75% and 79% viable embryos respectively) (Fig. 2b). As the expression of *cifA*$_{w\mathrm{AlbB(TIV)}}$ does not affect the fertility of females when crossed with wild-type males (Fig. 4b), the lower embryo viability rate in these crosses is likely due to the incomplete rescue of CifB$_{w\mathrm{AlbB(TIV)}}$-mediated inviability, rather than any CifA$_{w\mathrm{AlbB(TIV)}}$-mediated decrease in fertility.

We have shown that CifB-mediated toxicity is inhibited by CifA during spermatogenesis, consequently toxicity is likely to occur after CifB is deposited into oocytes upon fertilisation. Therefore, if rescue is dependent on CifA inhibition of CifB toxicity (through binding) then the levels of maternally deposited CifA must match or exceed that of paternally deposited CifB. It is probable that the levels of maternal CifA deposition (under the *exu* promoter) are not sufficient to prevent lethality in every embryo whereas maternally-inherited *w*AlbB *Wolbachia* likely produce CifA constantly which ensures CI rescue.

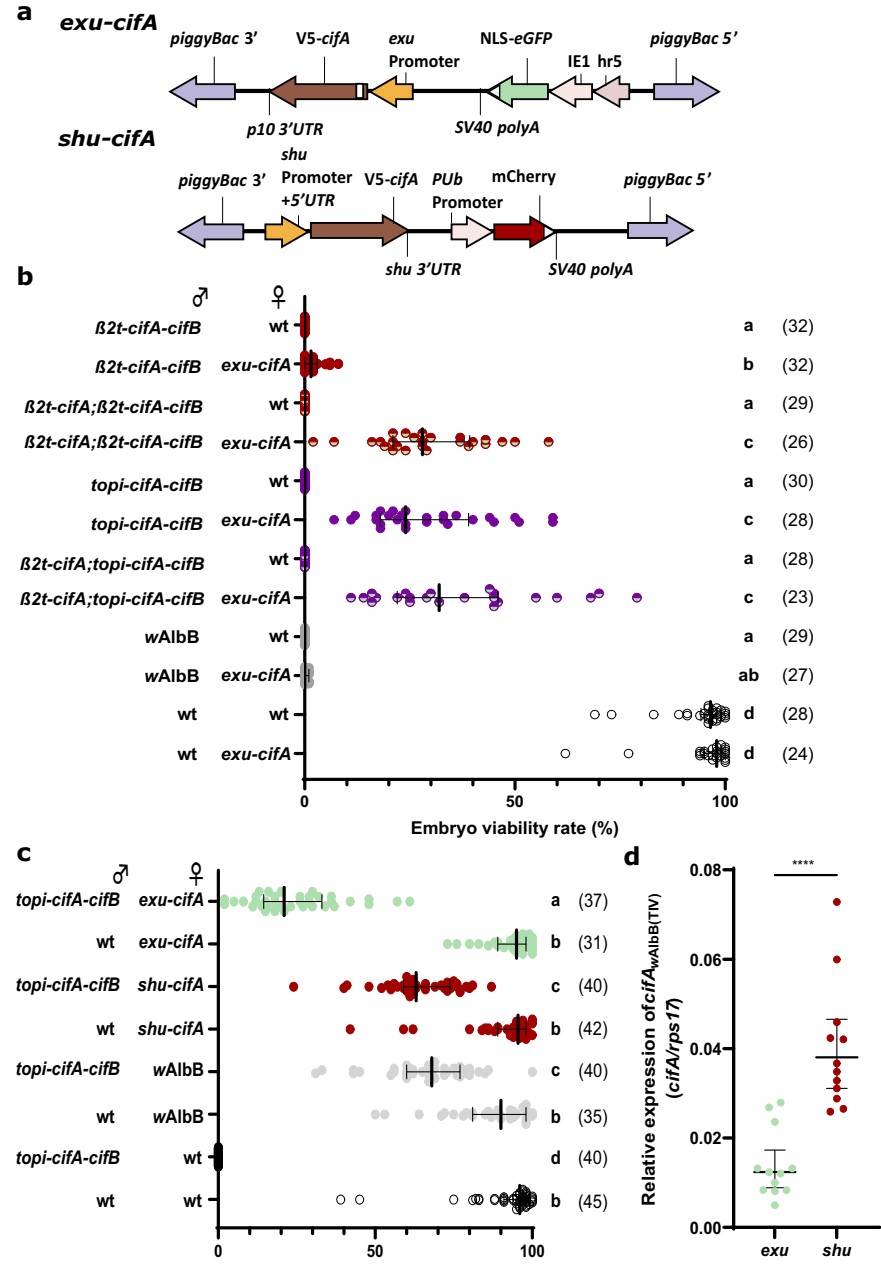

**Fig. 4 | Maternal *cifA*$_{w\text{AlbB(TIV)}}$ expression rescues CI induced by paternal *cifA*/*B*$_{w\text{AlbB(TIV)}}$ expression. a** Construct design of *exu-cifA* and *shu-cifA*. **b** Manipulating the paternal Cif dosage in transgenic males improves the capacity of *exu-cifA* females to rescue incompatibility. Expression of *cifA* in females does not affect their fertility when crossed with wild-type (wt) males. **c** Rescue capacity of transgenic females is increased when *cifA* expression is regulated by the *shu* promoter. In both **b** and **c**, Lines denote median and error bars interquartile ranges, numbers in parentheses denote the *n*. Letters indicate significant differences with an α = 0.05 calculated by a Kruskal–Wallis test: (**b** H = 311.0, *P* < 0.0001, d.f. = 11)(**c** H = 243.2, *P* < 0.0001, d.f. = 7) followed by a two-stage linear step-up procedure of Benjamini, Krieger, and Yekutieli to correct for multiple comparisons, individual *P*-values are listed in Source Data File. **d** The expression of *cifA* relative to *rps17* was higher in gravid ovaries taken from *shu-cifA* compared to *exu-cifA* females (two-tailed Mann–Whitney U test *P* < 0.0001, mean and s.d are shown, *n* = 12 for both groups). Source data are provided as a Source Data file.

Consequently, to improve the rescue capacity of the transgenic females an equal or greater abundance of CifA relative to CifB must be achieved, which may be generated through the lowering of paternal CifB or the increasing of maternal CifA deposition. An overexpression of *cifA*$_{w\text{Pip(TI)}}$ in transgenic *Anopheles gambiae* males was found to reduce the penetrance of CI induced by *cifA/B*$_{w\text{Pip(TI)}}$ co-expression[11], suggesting that significantly higher levels of *cifA* expression reduces the deposition of unbound CifB into fertilised oocytes. To test whether an overexpression of *cifA*$_{w\text{AlbB(TIV)}}$ in males might reduce the deposition of unbound CifB$_{w\text{AlbB(TIV)}}$, and therefore improve the rescue capability of transgenic females, males heterozygous for both the *β2t-cifA*$_{w\text{AlbB(TIV)}}$ (A2) and *topi-cifA*$_{w\text{AlbB(TIV)}}$-*cifB*$_{w\text{AlbB(TIV)}}$ insertions were crossed with *exu-cifA*$_{w\text{AlbB(TIV)}}$ females. A higher expression of *cifA*$_{w\text{AlbB(TIV)}}$ comparative to *cifB*$_{w\text{AlbB(TIV)}}$ in *β2t-cifA*$_{w\text{AlbB(TIV)}}$;*topi-cifA*$_{w\text{AlbB(TIV)}}$-*cifB*$_{w\text{AlbB(TIV)}}$ males did not significantly increase rescue capability of either wild-type or *exu-cifA*$_{w\text{AlbB(TIV)}}$ females (Fig. 4b). This suggests that the increase in embryo viability rate observed for *β2t-cifA*$_{w\text{AlbB(TIV)}}$;*β2t-cifA*$_{w\text{AlbB(TIV)}}$-*cifB*$_{w\text{AlbB(TIV)}}$ in comparison to

$\beta2t$-$cifA_{w\mathrm{AlbB(TIV)}}$-$cifB_{w\mathrm{AlbB(TIV)}}$ crosses (Figs. 2b and 4b) is the result of an inhibition of CifB$_{w\mathrm{AlbB(TIV)}}$-induced toxicity in the male germline, and not a reduction in deposition of the toxin.

The promoter/3'UTR sequences of the *Ae. aegypti shut-down (shu)* gene were expected to improve deposition into developing oocytes[38,39]. Therefore, a *shu*-$cifA_{w\mathrm{AlbB(TIV)}}$ line (Fig. 4a) was generated and heterozygous females were crossed with *topi*-$cifA_{w\mathrm{AlbB(TIV)}}$-$cifB_{w\mathrm{AlbB(TIV)}}$ males (Fig. 4c). Changing the regulatory sequences resulted in a 42% increase in embryo viability in comparison to *exu*-$cifA_{w\mathrm{AlbB(TIV)}}$ females when mated with transgenic males, with *shu*-$cifA_{w\mathrm{AlbB(TIV)}}$ females being able to rescue incompatibility to a level comparable to those carrying *w*AlbB *Wolbachia* (median of 63% and 68% viable embryos respectively) (Fig. 4c). This increase coincided with a significantly higher $cifA_{w\mathrm{AlbB(TIV)}}$ expression in blood-fed *shu*-$cifA_{w\mathrm{AlbB(TIV)}}$ ovaries (Fig. 4d). However, different expression patterns between the promoters might also result in different levels of CifA deposition.

## Discussion

The capacity to mediate cytoplasmic incompatibility is central to the remarkable ecological success of *Wolbachia* and is key to the deployment of *Wolbachia* as a vector control tool. A variety of artificial *Wolbachia* transinfections using different strains have now been generated in *Ae. aegypti*, and all - with the exception of *w*Au, which does not encode an active *cif* gene pair – induce highly penetrant CI[21–24,40]. *Ae. aegypti* is an ideal model organism for investigating *cif* gene function given its very high CI penetrance. This may be due in part to the fact that *Ae. aegypti* is not a native *Wolbachia* carrier, whereas in *D. melanogaster* CI has incomplete penetrance probably due to compensatory host adaptations aimed at suppressing CI and thereby minimising associated fitness costs[41,42]. Furthermore, the severe and growing public health burden of *Ae. aegypti*, its tractability in the laboratory, and the potential for building *cif* gene-based drive systems that require relatively high transgene population frequencies before drive is achieved (high-threshold drive)[43], makes studies on *cif* gene function in this species particularly timely and relevant. Although considerable research on the effects of *cif* gene expression using transgenic tools has been conducted in *D. melanogaster*[1,2,6–10,12], the synthetic recapitulation of CI using the *cif* genes in *Aedes* mosquitoes has not yet been reported. Here we demonstrate transgenic CI induction and rescue in this species using a previously untested Type IV *cif* gene pair from the *w*AlbB *Wolbachia* strain.

As both $cifA_{w\mathrm{AlbB(TIV)}}$ and $cifB_{w\mathrm{AlbB(TIV)}}$ expression was necessary for CI induction whilst only $cifA_{w\mathrm{AlbB(TIV)}}$ was required for rescue, our results conform to the previously described two-by-one model of CI[9]. However, this model does not elucidate the role(s) of paternal *cifA* expression in the induction mechanism. Here, we provide clear evidence for the role of CifA in the attenuation of CifB-mediated toxicity in the male germline. CifA-mediated inhibition of CifB toxicity has been well documented in both transgenic yeast and insect cell studies and is largely attributed to the formation of CifA-CifB heterodimers, which prevents CifB from directly interacting with unknown chromatin targets[1,7,8,12,17]. The requirement for the inactivation of the CifB$_{w\mathrm{AlbB(TIV)}}$ toxin in the testes to generate rescuable sterility in our study indicates that the modification/toxicity associated with sterility induction does not occur in the male reproductive tissues, but rather suggests it occurs in the fertilised embryo where *Wolbachia*-derived or maternally deposited CifA$_{w\mathrm{AlbB(TIV)}}$ is required to prevent sperm-bound CifB$_{w\mathrm{AlbB(TIV)}}$ toxicity (Supplementary Fig. 2). When binding is prevented through the mutation of residues in the CifA-CifB binding interface, transgenic *D. melanogaster* females which express the mutant *cifA* are unable to rescue CI induced by *cifA*-*cifB* expressing males[12], suggesting that the same mode of toxin inhibition is maintained both during spermatogenesis and after fertilisation in compatible *Wolbachia* crosses.

Although we demonstrated the importance of CifA-mediated inhibition of CifB toxicity during spermatogenesis in this study, there have been reports of CI induction through the expression of *cifB* alone[10,11], which suggests that paternal CifA is not always essential. It is probable that differences in i) *cifB* expression levels, ii) host sensitivity to toxicity, and iii) differences in catalytic activity between *cifB* homologues contribute to variations between studies. Interestingly, several studies have also demonstrated that *cifB* expression does not affect male fertility unless *cifA* is also expressed[2,6,9]. One explanation for this phenomenon is that in addition to its role in toxin inhibition there is evidence to suggest that CifA:CifB complex formation might suppress the premature degradation of CifB[10,17]. Therefore, CifA might be necessary to both inhibit CifB toxicity and ensure the correct deposition of CifB into sperm nuclei. Alternatively, another study has shown that paternal $cifB_{w\mathrm{Mel(TI)}}$ expression resulted in spermatid DNA damage, yet the fertility of these males was unaffected[18] – which might suggest that *cifB* expression could select for undamaged sperm that receive lower levels of CifB.

The transgenic recapitulation of CI induction and rescue in our study opens the potential for the generation of gene drive systems capable of spreading broad-action anti-pathogen effectors (linked to the *cif* drive elements) through targeted *Ae. aegypti* populations. A simple way to build a *cif* drive system using promoter sequences already outlined in this study would be to have *cifA* and *cifB* in the same transgene construct and introduced into a single genomic locus. The drive system could be constructed with the *topi* promoter driving male-specific germline expression of *cifA* and *cifB* and *shu* driving female germline expression of *cifA*. Provided that a critical population threshold frequency is surpassed, the advantage of carrying the *cif* drive allele for females (conferring CI rescue in matings with *cif* drive allele carrying males) will outweigh the cost incurred by carrying the *cif* drive allele for males (causing incompatibility in matings with wild-type females), and the *cif* drive system would be expected to spread and persist. Mathematical modelling of drive systems based on the *cif* genes in various design configurations suggests that they will possess high invasion-threshold frequencies, and thus are likely to be inherently more confinable to target populations than analogous homing drive systems; by splitting the induction and rescue components the drive could be further confined temporally as well as spatially[43]. Although requiring higher population frequencies before drive is achieved and therefore necessitating a greater volume of initial mosquito releases, these drive designs are expected to be more easily controlled and regulated from a biosafety and geopolitical perspective. While *Wolbachia* virus inhibition and density have so far remained stable in field populations of *Ae. aegypti* over several years, it is possible that in the longer term both will be ameliorated by natural selection, and likewise it is possible that "escape" mutations will arise in dengue or other arboviruses that reduce the effectiveness of transmission blocking. In this context it is important to continue research aiming to develop transgenic approaches for spreading antiviral effectors in this species. In addition, not all disease-transmitting insect species are likely to be able to support *Wolbachia* transinfections. Meanwhile, the sterility generated by *cifB*-expressing transgenic males provides additional options for innunaditive sterile male-only release strategies, aimed at population suppression of *Ae. aegypti* or other pest insects. Moreover, as we have shown that the presence of CifB can cause cell damage, it is possible that some of the fitness costs associated with the presence of some *Wolbachia* strains (e.g. reduced lifespan, fecundity, egg hatch rates) may be in part due to cumulative DNA damage resulting from unbound CifB. If this is the case, transgenic expression of the cognate CifA under a non-specific promoter may mitigate this damage and could result in *Wolbachia* transinfections with improved fitness profiles.

## Methods

### Generation of constructs

The coding sequences of the Type IV $w$AlbB $cifA$ (QBB83746.1) and $cifB$ (QBB83745.1) genes were codon optimised for expression in *Ae. aegypti*, synthesised, and cloned into pUC-GW-Amp plasmids using the GENEWIZ PriorityGENE service (Azenta Life Sciences). Constructs were engineered to express the *cif* genes either together (linked with a T2A peptide sequence) or independently under the control of the germline-specific promoters *β2t*, *topi*, *exu*, or *shu*. Fluorescent marker genes were regulated by either the Hr5/ie1 promoter/enhancer or *PUb* promoter sequences. The complete sequences of the $β2t\text{-}cifB_{w\text{AlbB(TIV)}}$, $β2t\text{-}cifA_{w\text{AlbB(TIV)}}$, $β2t\text{-}cifA_{w\text{AlbB(TIV)}}\text{-}cifB_{w\text{AlbB(TIV)}}$, $topi\text{-}cifA_{w\text{AlbB(TIV)}}\text{-}cifB_{w\text{AlbB(TIV)}}$, $exu\text{-}cifA_{w\text{AlbB(TIV)}}$, and $shu\text{-}cifA_{w\text{AlbB(TIV)}}$ plasmids have been deposited in GenBank with the accession numbers OR961086, OR961087, OR961088, OR961089, OR961090, and OR961091 respectively.

### Mosquito maintenance and experimental procedures

**Mosquito rearing.** All mosquito colonies were maintained at 27 °C and 70% humidity with 12-h light/dark cycles. Larvae were fed tropical fish pellets (Tetra) whilst adults had access to 5% sucrose solution *ad libitum*. Blood meals were provided using a Hemotek artificial blood-feeding system (Hemotek Ltd) and human blood (Scottish National Blood Bank). Damp Grade 1 filter-paper (Whatman plc) was provided as an oviposition source for egg collection. Eggs were desiccated for 5–10 days before hatching in water containing 1 g/l bovine liver powder (MP Biomedicals). The $w$AlbB-carrying *Ae. aegypti* line used in this study was generated previously through embryo microinjection[40].

**Generation of transgenic lines.** The posterior pole of freshly laid pre-blastoderm *Ae. aegypti* embryos (<2 h old) were microinjected[44] using a Nikon Eclipse TS100 microscope and air pump (Jun-Air). The injection mix consisted of a final concentration of 500 ng/μl donor plasmid and 300 ng/μl helper plasmid [PUb hyperactive *piggyBac* transposase (AGG1245)][38] in 1x injection buffer. Microinjection survivors ($G_0$) were screened for marker gene expression at the 4th instar larval stage using the Leica M165FC fluorescent microscope and appropriate filter setting, those transiently expressing the marker gene were individualised and mated with 3 wild-type mosquitoes. To identify transgenic lines the resultant $G_1$ generation from these crosses were likewise screened at the 4th instar. Lines were maintained by mating virgin transgenic females with an excess of wild type males, offspring were screened at 4th instar stage.

**Embryo viability assays.** Virgin females were mated with an excess of males in small (15 × 15 × 15 cm) Bugdorm cages (Megaview Science Co.). After blood-feeding, non-bloodfed females were removed and remaining individuals were left 3 days to become gravid. Females were individualised onto small damp Grade 1 filter-paper disks (Whatman plc). Females were left for 2 days to lay eggs, after which the females were removed and spermathecae were dissected to confirm mating. Egg cones were left for 5–10 days before counting the percentage viability of embryos on each cone. The viability (ie. the ability to hatch) of *Ae. aegypti* embryos can be observed easily due to phenotypic differences using a light microscope - inviable embryos display egg collapse which signals an early arrest in development, whilst viable embryos remain turgid indicating embryo development. To ensure the percentage viability count was reliable selected cones were floated, and after 2 days the egg cones were removed and dried, and the percentage hatch rate was calculated.

**RNA extraction and reverse transcription quantitative PCR (RT-qPCR) analyses.** Male reproductive tissues were dissected in pools of 3, homogenised in TRIzol reagent (Thermo Fisher Scientific) and stored at −80 °C. Ovaries were dissected from individual females 72 h after blood-feeding. RNA extractions were conducted as instructed by the manufacturer's instructions and the pelleted RNA was resuspended in RNase free water. The purity /concentration of RNA was quantified using the NanoDrop One spectrophotometer (Thermo Fisher Scientific) and up to 2 μg of RNA was used per 20 μl cDNA synthesis reaction (High-Capacity cDNA Reverse Transcription Kit, Thermo Fisher Scientific). 10 μl RT-qPCR reactions consisting of 2 μl cDNA, 5 μl Fast SYBR™ Green Master Mix (Thermo Fisher Scientific), 2 μl distilled water, and 0.5 μl of both forward and reverse primers (5 μM) were run on the QuantStudio™ 3 Real-Time PCR System (Thermo Fisher Scientific). The primers used in this study were: cifAF, 5' GCGAACGATACACCACCTTC 3'; cifAR, 5' TTCCCACACGTT CATCATGC 3'; cifBF, 5' AAGATCGCCATCCTGACCAA 3'; cifBR, 5' GCGATTTTCTCCAGCTCTCC 3'; rps17F, 5' CACTCCCAGGTCCGTGGT AT 3'; rps17R, 5' GGACACTTCCGGCACGTAGT 3'. The primers targeting *cifB* could be used to determine the relative expression of either *cifB* alone or the bicistronic *cifA-cifB* sequence. The $2^{-ΔCt}$ method was used to determine the expression of the *cif* genes relative to the housekeeping reference gene *rps17*.

### Microscopy

**Brightfield microscopy.** The reproductive tissues (testes, seminal vesicle and accessory glands) from 5-day old males were dissected in PBS and mounted onto a glass slide. The coverslip was lightly pressed to break the tissue and release sperm (if any was produced). A series of brightfield images were taken using the Leica DMi8 (Leica Microsystems) and 10× objective lens, successive images were used to create time-lapse videos.

**TUNEL assays.** Terminal deoxynucleotidyl transferase-mediated dUTP nick-end labelling (TUNEL) assays were used to detect single- and double-stranded DNA nicks and fragmentation via the ApopTag® Fluorescein Direct In Situ Apoptosis Detection Kit (Merck Group). Testes dissected from 5-day old males were fixed in 4% formaldehyde, permeabilised in PBST, and treated with components from the kit. For the positive control, a selection of wild-type testes were treated with DNase I (Thermo Fisher Scientific) for 20 min before fixation. Tissues were mounted in Fluoroshield™ with DAPI, histology mounting medium (Sigma-Aldrich) and imaged using a Zeiss LSM 880 confocal microscope (Zeiss) with a 20x objective. Fluorescein-labelled DNA breaks and DAPI-stained nuclei were imaged using a 488 nm and 405 nm laser respectively with GaAsP detectors, settings were kept constant for all images taken.

**Statistics and reproducibility.** Statistical analysis was performed using the GraphPad Prism software. A Shapiro−Wilk normality test was used to determine if the data was normally distributed. For multiple comparisons of embryo viability one or more of the groups were not normally distributed so non-parametric Kruskal−Wallis tests with were performed with a two-stage step-up procedure of the Benjamini, Krieger and Yekutieli method for controlling the false discovery rate. For comparisons of gene expression between two groups, when the data was normally distributed an unpaired *t*-test (two-tailed) was performed, if not normally distributed a Mann−Whitney U test was performed. For the TUNEL assays, a representative image of a dissected and stained teste from each group was selected from 4-10 independent tissue images. All data used for statistical analysis along with the results of the tests used are included in the Source Data file.

### Reporting summary

Further information on research design is available in the Nature Portfolio Reporting Summary linked to this article.

## Data availability

Datasets generated and/or analysed during the current study are available in the main manuscript or are appended as supplementary data. The source data, statistical analysis tests, and their outcomes used for figures are available to view in the Source Data file. The plasmid DNA sequences used to generate transgenic lines in this study are deposited in GenBank with the accession codes OR961086, OR961087, OR961088, OR961089, OR961090, and OR961091. Source data are provided with this paper.

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

## Acknowledgements

The study was supported by MRC award MC_ST_CVR_2019 and Wellcome Trust Award 202888/Z/16/Z.

## Author contributions

C.J.M., T.H.A., and S.P.S. conceived and designed the experiments. C.J.M. and T.H.A. collected, interpreted, and analysed the data. T.H.-S. and J.M. contributed with data interpretation. T.H.-S. and L.A. provided materials and H.W.-C. provided technical advice required for experimentation. C.J.M., T.A. and S.P.S. wrote the manuscript whilst all authors provided comments/edits and approved the final submission draft.

## Competing interests

The authors declare no competing interests.
