## [Peer Review File · Nature Communications]

REVIEWER COMMENTS

Reviewer #1 (Remarks to the Author):

dear Editor,

I'm writing in regard to the paper submitted by Cameron McNamera and colleagues, entitled "Transgenic expression of cif genes from Wolbachia strain wAlbB recapitulates cytoplasmic incompatibility in *Aedes aegypti*." Wolbachia bacteria are known to drive their spread through nature in part by inducing host manipulations such as Cytoplasmic Incompatibility (CI), which kills off the embryos of uninfected females. The current view is that Wolbachia proteins called Cytoplasmic Incompatibility Factors (Cifs), termed CifA and CifB appear to induce CI lethality through effects on developing sperm, whereas maternal CifA is able to confer "Rescue" of embryo viability.

The authors of this study investigated the *in vivo* functions of CifA and CifB (from wAlbB) by creating transgenic *Aedes aegypti* mosquitoes that express the AlbB transgenes in germline tissues. It has been unclear for some time to what extent the endonuclease domain of CifB is important for CI. The authors demonstrated that overexpressed paternal CifB can cause DNA damage during spermatogenesis, but paternal DNA damage is not required to induce CI. Among other interesting results, the authors also convincingly showed a requirement for CifA to counteract paternal CifB function, during spermatogenesis as well as post-fertilization. These findings in themselves reframe the definition of what "Rescue" actually means.

Overall I found this to be a very strong paper, with robust attention to experimental design, inclusion of appropriate controls, and clarity of communication in this well-written manuscript. I especially liked the authors' taking time to distinguish the basis for embryonic lethality, parsing the effects of defective spermatogenesis vs. lethality post-fertilization. It has been a significant detriment to this area of study that most papers have not made this distinction, and McNamera et al committing this attention to detail is commendable, and raises this study to a distinctive level of quality.

My minor comments are as follows: the only thing I would have liked to see but didn't would be inclusion of an endonuclease-mutant CifB transgene to show whether Cif-B induced DNA damage can be more generally excluded as the basis for CI *in vivo*, post-fertilization as well. (Am saying this only as there could still be other ways for CifB to induce damage, even without an endonuclease domain.) If this point has already been addressed by literature, the authors kindly including that in the Discussion would help. Also, I wasn't sure exactly what was intended with discussion of toxin deposition at the top-middle of page 7-- this was the only part of the paper that came through as a little unclear.

Reviewer #2 (Remarks to the Author):

This is a well-written article containing high quality science that is important for the field.

It looks to recapitulate cytoplasmic incompatibility, a form of reproductive manipulation performed by Wolbachia bacteria in some insects, through the transgenic expression of cif genes that determine this behaviour. While the goal of transgenic CI is not entirely novel and has been performed in other insects and some mosquitoes, this article is still a significant advance for the field in that: 1- it has achieved some considerable success in doing so, and getting there was not trivial 2- perhaps more importantly, it sheds light on competing theories for the molecular mechanism of CI and the mode of interaction between the CifA and CifB genes. These results strongly support the existence of a toxin:antidote system and go a long way to deciphering when and where this happens, and how to modulate this experimentally

The process of experimentation is described very logically and the conclusions are sound. Where there are potential alternative explanations for findings, these are well caveated.

This work will be important for future attempts at turning this system into a working gene drive.

I have some minor comments and suggestions that mostly go towards readability (though I accept it is a hard and complicated subject area to convey to a very wide readership). Rather than list all of these here I have attached an annotated copy of the manuscript.

Worthy of note here:

what is missing in the Discussion of what are the next steps to make this into a gene drive? How feasible? How far away are we with these current strains? Unknowns etc.? etc

Surely a cartoon figure of competing models would be helpful?

I think it would be much easier if more emphasis could be made in distinguishing sources of CifA, B etc from those provided by the bacteria.

Quite a few terms would need a little more explanation for a non-specialist (see text)

Could the transgenic expression of some of these components aid in colonising new strains of Wolbachia, by making them more permissive? As a general strategy I mean? Is that not worth mentioning, if so?

Reviewer 1

the only thing I would have liked to see but didn't would be inclusion of an endonuclease-mutant CifB transgene to show whether Cif-B induced DNA damage can be more generally excluded as the basis for CI in vivo, post-fertilization as well. (Am saying this only as there could still be other ways for CifB to induce damage, even without an endonuclease domain.) If this point has already been addressed by literature, the authors kindly including that in the Discussion would help.

In the original manuscript (Lines 126-128) we acknowledged the caveat that the TUNEL assay “does not discriminate whether the DNA damage observed is the direct result of CifB_{wAlbB(TIV)} nuclease activity or whether CifB_{wAlbB(TIV)} induces apoptosis through another mechanism. However, previously published research has shown that mutations in the nuclease domains inhibits toxicity in yeast and reduces CI penetrance in transgenic fly studies. We have added the following statement to reference these studies:

The toxicity of CifB peptides encoding functional nuclease domains (Types II-V) is likely to be dependent on nuclease catalytic activity; mutation of key catalytic residues in these domains ablates toxicity in transgenic yeast^{1,7,10} and reduces CI penetrance in *Drosophila*⁷. However, it is not known if this activity occurs in the male reproductive tissues.

Also, I wasn't sure exactly what was intended with discussion of toxin deposition at the top-middle of page 7-- this was the only part of the paper that came through as a little unclear.

This section has now been re-written and expanded to improve clarity. The section now reads: “We have shown that CifB-mediated toxicity is inhibited by CifA during spermatogenesis, consequently toxicity is likely to occur after CifB is deposited into oocytes upon fertilisation. Therefore, if rescue is dependent on CifA inhibition of CifB toxicity (through binding) then the levels of maternally deposited CifA must match or exceed that of paternally deposited CifB. It is probable that the levels of maternal CifA deposition (under the *exu* promoter) are not sufficient to prevent lethality in every embryo whereas maternally-inherited *wAlbB Wolbachia* likely produce CifA constantly which ensures CI rescue. Consequently, to improve the rescue capacity of the transgenic females an equal or greater abundance of CifA relative to CifB must be achieved, which may be generated through the lowering of paternal CifB or the increasing of maternal CifA deposition. An overexpression of *cifA*_{wPip(TI)} in transgenic *Anopheles gambiae* males was found to reduce the penetrance of CI induced by *cifA/B*_{wPip(TI)} co-expression¹¹, suggesting that significantly higher levels of *cifA* expression reduces the deposition of unbound CifB into fertilised oocytes. To test whether an overexpression of *cifA*_{wAlbB(TIV)} in males might reduce the deposition of unbound CifB_{wAlbB(TIV)}, and therefore improve the rescue capability of transgenic females, males heterozygous for both the *β2t-cifA*_{wAlbB(TIV)} (A2) and *topi-cifA*_{wAlbB(TIV)}-*cifB*_{wAlbB(TIV)} insertions were crossed with *exu-cifA*_{wAlbB(TIV)} females. A higher expression of *cifA*_{wAlbB(TIV)} comparative to *cifB*_{wAlbB(TIV)} in *β2t-cifA*_{wAlbB(TIV)};*topi-cifA*_{wAlbB(TIV)}-*cifB*_{wAlbB(TIV)} males did not significantly increase rescue capability of either wild-type or *exu-cifA*_{wAlbB(TIV)} females (Figure 4b). This suggests that the increase in embryo viability rate observed for *β2t-cifA*_{wAlbB(TIV)};*β2t-cifA*_{wAlbB(TIV)}-*cifB*_{wAlbB(TIV)} in comparison to *β2t-cifA*_{wAlbB(TIV)}-*cifB*_{wAlbB(TIV)} crosses (Fig. 2b and Fig. 4b) is the result of an inhibition of CifB_{wAlbB(TIV)}-induced toxicity in the male germline, and not a reduction in deposition of the toxin. “

Reviewer 2

what is missing in the Discussion of what are the next steps to make this into a gene drive? How feasible? How far away are we with these current strains? Unknowns etc.? etc

A description of a potential *cif* gene drive configuration has now been added to the Discussion. Reference has been made to the fact that the configuration outlined uses *cif* genes and promoter sequences already characterised in the ms. The additional sentences read:

A simple way to build a *cif* drive system using promoter sequences already outlined in this study would be to have *cifA* and *cifB* in the same transgene construct and introduced into a single genomic locus. The drive system could be constructed with the *topi* promoter driving male-specific germline expression of *cifA* and *cifB* and *shu* driving female germline expression of *cifA*. Provided that a critical population threshold frequency is surpassed, the advantage of carrying the *cif* drive allele for females (conferring rescue in matings with *cif* drive allele-carrying males) will outweigh the cost incurred by carrying the *cif* drive allele for males (causing incompatibility in matings with wild-type females), and the *cif* drive system would be expected to spread and persist.

Surely a cartoon figure of competing models would be helpful?

A schematic has been added as a supplementary Figure. So as not to be overly complex and to be consistent with the style of a research article, we have focused the scheme on results presented in the manuscript only; that is, on the effects of the relative abundances of wAlbB *cifA* and *cifB*.

I think it would be much easier if more emphasis could be made in distinguishing sources of CifA, B etc from those provided by the bacteria.

This has now been clarified throughout the text. It is now clearly stated when crosses involved *Wolbachia*-carrying females.

Quite a few terms would need a little more explanation for a non-specialist (see text)

We have made the following amendments:

Line 44: co-expression has been clarified and now reads: *However, for some host species and/or cif homolog combinations, the simultaneous co-expression of both cifA and cifB was found to be necessary for CI induction.*

Line 69: the transmission blocking effect has been expanded on and now reads: *Wolbachia strains introduced into Ae. aegypti from other insect host species have resulted in the generation of Ae. aegypti lines that exhibit strong CI²¹⁻²⁴ and block the transmission of some positive-strand RNA viruses of significant public health concern, such as dengue virus.*

Line 77: Included information of the differences/similarities between Type III and IV *cif* genes: *The wAlbB genome encodes two sets of cognate cif genes of different types (Type III and IV)³⁴. Despite significant sequence divergence, both sets possess the same putative protein domains and are assumed to be active since they do not contain predicted loss-of-function mutations; however, to date there have been no functional studies on them.*

Line 85: the principle of a gene-drive system is briefly described, and now reads: *The findings of this study provide insights into the role of CifA in the male germline and lays the groundwork for the construction of a transgenic system capable of inheritance at greater than Mendelian ratios; such a synthetic cif gene drive system could be used for the delivery of antiviral effectors in Ae. aegypti.*

Line 88: The nomenclature has now been added to Line 83.

Line 106: 'rate' has been changed to 'frequency'.

Line 117: Percentage median embryo viability rate has been included.

Line 148: Percentage median embryo viability rate has been included.

Line 164: text now clarified here (and elsewhere) to indicate that *wAlbB* refers to *Wolbachia* bacteria.

Line 166: header now amended to read: 'Maternal transgenic expression of *cifA*_{wAlbB(TIV)} rescues transgenic *CifB*_{wAlbB(TIV)}-induced sterility'.

Line 177: text amended to include the word 'partial'.

Line 184: text now amended to read: *the lower embryo viability rate in these crosses is likely due to the incomplete rescue of *CifB*_{wAlbB(TIV)}-mediated inviability, rather than any *CifA*_{wAlbB(TIV)}-mediated decrease in fertility.*

Line 190: The word unbound was included to allow for different interpretation of the results.

Line 191: As stated in response to reviewer 1 this section has been reworded to help the reader follow the logic of the experiment.

Line 218: Explanation of high-threshold gene drives has been added, now reads: the potential for building *cif* gene-based drive systems that require relatively high transgene population frequencies before drive is achieved (high-threshold drive)⁴⁰

Line 239-249: A schematic listed as Supplementary Figure 2 was created, the reasoning behind focussing on our results only was stated in the response to "Surely a cartoon figure of competing models would be helpful?"

Line 252: (linked to the *cif* drive elements) was added

Line 253: was addressed in the initial comment "what is missing in the Discussion of what are the next steps to make this into a gene drive? How feasible? How far away are we with these current strains? Unknowns etc.? etc"

Line 258: Introduced why the limitation of high-threshold gene drives (a higher introduction frequency/less penetrant drive) is actually a benefit when it comes to managing gene drive spreads after release. Now reads: Although requiring higher population frequencies before drive is achieved and therefore necessitating a greater volume of initial mosquito releases, these drive designs are expected to be more easily controlled and regulated from a biosafety and geopolitical perspective.

Line 266: Now reads: innunaditive sterile male-only release strategies

Could the transgenic expression of some of these components aid in colonising new strains of *Wolbachia*, by making them more permissive? As a general strategy I mean? Is that not worth mentioning, if so?

The discussion has now been amended to include the following sentences:

Moreover, as we have shown that the presence of *CifB* can cause cell damage, it is possible that some of the fitness costs associated with the presence of some *Wolbachia* strains (e.g. reduced lifespan, fecundity, egg hatch rates) may be in part due to cumulative DNA damage resulting from unbound *CifB*. If this is the case, transgenic expression of the cognate *CifA* under a non-specific promoter may mitigate this damage and could result in *Wolbachia* transinfections with improved fitness profiles.

REVIEWERS' COMMENTS

Reviewer #1 (Remarks to the Author):

The authors response has fully addressed my review.

Reviewer #2 (Remarks to the Author):

Sound. I would say this but the manuscript is even better, and certainly more readable and accessible, after incorporating reviewer's suggestions.